# Yeast Chromatin Mutants Reveal Altered mtDNA Copy Number and Impaired Mitochondrial Membrane Potential

**DOI:** 10.3390/jof9030329

**Published:** 2023-03-07

**Authors:** Dessislava Staneva, Bela Vasileva, Petar Podlesniy, George Miloshev, Milena Georgieva

**Affiliations:** 1Laboratory of Molecular Genetics, Epigenetics and Longevity, Institute of Molecular Biology “RoumenTsanev”, Bulgarian Academy of Sciences, 1113 Sofia, Bulgaria; 2CiberNed (Centro Investigacion Biomedica en Red Enfermedades Neurodegenerativas), 28029 Barcelona, Spain

**Keywords:** chromatin, *hho1*
*Δ*, *arp4*, ageing, mitochondria, rho- phenotype, mtDNA, mitochondrial membrane potential, chronological lifespan

## Abstract

Mitochondria are multifunctional, dynamic organelles important for stress response, cell longevity, ageing and death. Although the mitochondrion has its genome, nuclear-encoded proteins are essential in regulating mitochondria biogenesis, morphology, dynamics and function. Moreover, chromatin structure and epigenetic mechanisms govern the accessibility to DNA and control gene transcription, indirectly influencing nucleo-mitochondrial communications. Thus, they exert crucial functions in maintaining proper chromatin structure, cell morphology, gene expression, stress resistance and ageing. Here, we present our studies on the mtDNA copy number in *Saccharomyces cerevisiae* chromatin mutants and investigate the mitochondrial membrane potential throughout their lifespan. The mutants are *arp4* (with a point mutation in the *ARP4* gene, coding for actin-related protein 4—Arp4p), *hho1Δ* (lacking the *HHO1* gene, coding for the linker histone H1), and the double mutant *arp4 hho1Δ* cells with the two mutations. Our findings showed that the three chromatin mutants acquired strain-specific changes in the mtDNA copy number. Furthermore, we detected the disrupted mitochondrial membrane potential in their chronological lifespan. In addition, the expression of nuclear genes responsible for regulating mitochondria biogenesis and turnover was changed. The most pronounced were the alterations found in the double mutant *arp4 hho1Δ* strain, which appeared as the only petite colony-forming mutant, unable to grow on respiratory substrates and with partial depletion of the mitochondrial genome. The results suggest that in the studied chromatin mutants, *hho1Δ*, *arp4* and *arp4 hho1Δ*, the nucleus-mitochondria communication was disrupted, leading to impaired mitochondrial function and premature ageing phenotype in these mutants, especially in the double mutant.

## 1. Introduction

Mitochondria are multifunctional cellular organelles with an endosymbiotic origin that are crucially important for eukaryotic cells and are intrinsically connected with ageing [1,2,3]. Most of the present knowledge of how mitochondria work was initially discovered using *Saccharomyces cerevisiae,* starting with pioneer works about 60 years ago [4]. Moreover, the capability of yeast to use fermentation as an energy source allowed various mitochondrial dysfunction-causing mutations to be studied, giving it the aptitude to survive even without the presence of mitochondrial DNA (mtDNA) or damaged oxidative phosphorylation, as long as it was supplemented with a fermentable carbon source [5]. Studying mitochondrial dynamics in yeast has also potentiated the discovery of the association between human diseases and mitochondrial fusion and fission defects [6] and the identification of drugs that target mitochondrial dysfunction [7,8].

*S. cerevisiae* was used as a model organism for studying the link between the ageing process and mitochondrial function, which turned out to be tightly interconnected [9,10,11]. Data have shown accelerated yeast ageing due to mitochondrial fission reduction [12]. Daughter cell rejuvenation has been linked to the retention of damaged mitochondria by the mother cell [13,14]. Lack of mitochondrial function was detrimental to yeast chronological lifespan (CLS) [15]. The loss of mitochondrial membrane potential (MMP, denoted as ΔΨmt) proved essential for proper mitochondrial respiration in yeast [16] and, in turn, triggered genomic instability [17]. Importantly, alterations to mitochondrial function have been linked to triggering changes in the expression of nuclear genes responsible for regulating mitochondria biogenesis and turnover, thus allowing the cells to cope with cellular adaptation to the new conditions, a process called mitochondrial retrograde response [18]. For example, in *S. cerevisiae*, it was demonstrated that the main pathway of the mitochondrial retrograde response was the sensing of changes in ΔΨmt, thus initiating metabolism reconfiguration by changes in the expression of genes responsible for the fatty acid β-oxidation, peroxisomal biogenesis, and pathways supplying mitochondria with citrate and acetyl-CoA [19].

Interestingly, mitochondrial retrograde response activation has been proven in certain conditions to extend the yeast replicative lifespan (RLS) [20]. Woo and Poyton have discovered that an increase in RLS was also achieved in rho^0^ yeast cells that lack their mitochondrial DNA [21]. Being endosymbiotic organelles, mitochondria have multiple genome copies, compacted with mtDNA binding proteins forming nucleoids. Usually, *S. cerevisiae* has between 40 and 60 nucleoids, each containing 1 or 2 copies of mtDNA [22]. Even though mitochondria have their genome and translation machinery, the cell nucleus and mitochondria are still interdependent. This connection is represented by the fact that different mitochondrial proteins, for example, ATP synthase complex subunits, are encoded by nuclear genes and only after their maturation are transferred to the mitochondria [23]. mtDNA copy number alterations have been linked to several metabolic, neurodegenerative diseases and different types of cancer [24,25,26].

Chromatin structure influences mtDNA copy number, as improper chromatin assembly or reduction in core histones expression in *S. cerevisiae* have been shown to promote an increase in the mtDNA copy number, oxidative phosphorylation, ATP synthesis, oxygen consumption, and the expression of tricarboxylic acid cycle enzyme encoding genes [27]. Although there are data for evolutionary conserved pathways and mechanisms that control the mtDNA copy number [28], the profound knowledge of these mechanisms remains not fully understood.

In our previous studies, we have proven that the physical interaction between two crucial chromatin components—the yeast linker histone Hho1p, responsible for nucleosome stabilization and higher order chromatin structure maintenance [29,30,31], and the actin-related protein Arp4p—a subunit of Ino80, SWR1 and NuA4 [32], was of utmost importance for both yeast replicative and chronological ageing. The studied chromatin mutants were *arp4* (with a point mutation in the *ARP4* gene, coding for actin-related protein 4—Arp4p), *hho1Δ* (lacking the *HHO1* gene, coding for the linker histone H1), and the double mutant *arp4 hho1Δ* cells with the two mutations. We have shown that the disruption of the interaction between the linker histone and Arp4p affected the organization of chromatin structure, cellular morphology, and how cells responded to stress. Moreover, the double mutant cells that experienced premature ageing phenotypes had a reduced chronological and replicative lifespan and lowered replicative potential [33,34,35,36,37].

The present work shows that the studied chromatin *S. cerevisiae* mutants had strain-specific changes in the mtDNA copy number. Furthermore, we detected impaired mitochondrial membrane potential in their chronological lifespan under optimal and UVA/B stress conditions. In addition, the expression of nuclear genes responsible for regulating mitochondria biogenesis and turnover was changed. The most pronounced were the alterations in the double mutant *arp4 hho1Δ* strain, which appeared to be the only petite colony-forming mutant, unable to grow on respiratory substrates and with partial depletion of the mitochondrial genome.

## 2. Materials and Methods

### 2.1. Yeast Strains and Cultivation Conditions

*Saccharomyces cerevisiae* strains used in this study are listed in Table 1. Cells were grown in YPD medium (1% Yeast extract, 2% Peptone, 2% D-glucose) at 28 °C, with aeration. The ability of cells to assimilate non-fermentable carbon sources was determined by plating single colonies on YPGly (with 2% glycerol) and YPEth (with 2% ethanol) media. The solid medium contained 2% agar (*w/v*). For examination of *S. cerevisiae* wild type and mutant strains in the course of chronological lifespan (CLS), the yeast cultures were propagated in synthetic complete dextrose media, SCD (1.7% yeast nitrogen base, 2% dextrose and 20 μg/mL of supplements according to the particular strain requirements) [38]. Cell cultivation continued for nine days at 30 °C in a water bath shaker. Cell aliquots have been taken for further analyses at specific time points, namely on the 4th, 24th hours, 3rd or 9th day of culturing.

### 2.2. Replica Plating and Spot Tests

Yeast cultures were propagated in YPD medium at 30 °C overnight. Cells were diluted in sterile water, plated on solid YPD media and incubated at 28 °C for 72 h to allow single colonies formation. Using sterile velveteen, each master YPD plate with single colonies was plated in replicas successively on YPGly, YPEth and YPD agar plates. The ability of WT and mutant strains to utilise and grow on different C-sources was compared after three days of incubation at 28 °C.

Yeast cells from a single colony grown on a YPD plate were taken with a toothpick and resuspended in sterile water. Four 10-fold serial dilutions of each strain were replica-plated with a plater onto solid YPGly, YPEth and YPD media and incubated for three days before documentation of resultant growth.

### 2.3. UVA/B Irradiation

Aliquots of yeast cells cultivated in SCD medium were taken at the 4th, 24th, and 72nd h and on the 9th day of cultivation and irradiated with UVA/B light for 3 (460.8 mJ) or 30 min (4608 mJ). A 15 W Cleo lamp was the UVA/B source giving an energy dose of 2.56 mW/cm^2^. After irradiation, cells recovered at 28 °C in a water bath shaker for 90 min. Control group samples followed the same protocol except for UVA/B irradiation.

### 2.4. Total DNA and RNA Preparation and Primers Used in PCR Analyses

Total DNA and RNA were yielded using the Dire*Ct*Quant 100ST DNA/RNA/Protein Solubilisation Reagent with Stabilized Temperature Indicator (DireCtQuant, Lleida, Spain) according to the manufacturer’s protocol. The nucleic acids pool was used for direct analysis by conventional, quantitative PCR (qPCR) or digital droplet PCR (ddPCR).

Oligonucleotides used in the present study are described in Table 2. Target-specific PCR primers were designed utilising Primer-BLAST [40]. The specificity of primers was verified versus the complete sequence of the *S. cerevisiae* genome. Primers for qPCR and ddPCR were designed to meet a target annealing temperature of 60 °C and a maximum amplicon length of 110 bases. Primers for examination of mitochondrial genome integrity were selected to cover different regions along the mtDNA, and combinations between forward and reverse primers resulted in 17 primer pairs giving amplicons with a wide range of expected length (Table 2). All oligonucleotides used in PCR analyses were designed based on the complete genome sequence of *S. cerevisiae* S288C mitochondrion (NCBI Reference Sequence: NC_001224.1) or nuclear genome (*ATG18* NM_001179986.1; *CDC28* NM_001178508.3 and *ATP25* NM_001182598.1).

### 2.5. Conventional PCR and Real-Time Quantitative PCR (qPCR) Analyses

The nuclear gene *ATP25*, coding for a mitochondrial inner membrane protein [41], was used as positive control; the *ATG18* and *CDC28* single-copy genes were used as the internal control in the PCR analyses. PCR reactions were performed using SG qPCR Master Mix (EUR_X_^®^ Sp., Gdansk, Poland) in a Rotor-Gene™ 6000 Real-time PCR thermal cycler (Corbett Life Science; Qiagen, Hilden, Germany). Depending on the expected amplicon length, the amplified PCR fragments were subsequently electrophoretically analysed in 0.8–1.8% agarose gel. For studying the mtDNA integrity, PCR analyses were performed under the following conditions: 95 °C for 10 min; 45 cycles of 15 s at 95 °C, 25 s at 56 °C, and 120 s at 72 °C. Conditions of the qPCR for copy number assessment included initial denaturation at 95 °C for 10 min followed by 45 cycles of 15 s at 95 °C, 20 s at 60 °C and 20 s at 60 °C with the acquisition. The end products were subjected to melting curve analyses to verify the specificity and identity of the amplified DNA fragments. To this end, melting curve data were collected between 60 and 95 °C rising by 1 degree each step, waiting for 5 s before acquisition. The obtained data were analysed using the Rotor-Gene™ 6000 Series Software 1.7 (Qiagen, Hilden, Germany), and the relative quantitation of DNA copy number was calculated via the ΔΔCt method (2^−ΔΔCt^) [42]. The sample WT 4th h was applied as a calibrator in all qPCR experiments.

### 2.6. Absolute Quantification of mtDNA Copy Number and ATG18 and CDC28 Gene Expression by Droplet Digital PCR (ddPCR)

Selfie-dPCR using EvaGreen fluorescent DNA-binding dye was performed as described previously [43,44]. The forward primer 8F (5′-TGAAGCTGTACAACCTACCGA-3′) and the reverse primer 8R (5′-ACCTGCGATTAAGGCATGATGA-3′) used in ddPCR analysis targeted the mt*COX3* gene of *S. cerevisiae* mtDNA (Table 2). The forward and reverse primers targeting single-copy nuclear genes, *ATG18* and *CDC28,* are described in Table 2.

To prepare DNA templates for copy number quantification by ddPCR, the obtained 100ST total DNA/RNA extracts were digested with restriction enzyme FastDigest BsuRI (*Hae*III) for mtDNA and FastDigest SaqAI (*Mse*I) for *ATG18* and *CDC28* genes for 15 min at 37 °C in ddPCR reaction digestion buffer (Thermo Scientific™, Thermo Fisher Scientific, Waltham, MA, USA). The reaction mix contained 10 μL 2X QX200™ ddPCREvaGreen Supermix (Bio-Rad catalogue number 186-4033, Hercules, CA, USA); 125 nM each of a forward and a reverse primer, the appropriate sample (100ST digested DNA/RNA) and QX200 Droplet Generation Oil for EvaGreen (Bio-Rad catalogue number 186-4005, Hercules, CA, USA). Conditions for ddPCR amplification were 5 min at 95 °C, 40 cycles of 30 s at 95 °C, 1 min at 60 °C, followed by 5 min at 4 °C and 90 °C for 5 min. The data were analysed using QuantaSoft™ v1.7 software (Bio-Rad, Hercules, CA, USA). First, the absolute number of mtDNA copies was calculated by dividing the detected mtDNA copy number by the mean number of *ATG18* and *CDC28* copies. This represented the absolute mtDNA copy number per haploid genome.

The absolute number of gene transcripts was calculated by subtraction from the copies measured in the (RT+) reaction and the copies measured in the (RT−) reaction. The resulting number (absolute number of transcripts) was divided by the number of the measured gene (RT−). The resulting measurement reflects the absolute number of transcripts encoded by the target gene. These results are not dependent on the expression level of a reference gene, the amount of the sample or genomic differences between the strains [43].

### 2.7. FACS Analysis for Rhodamine 123 Staining

For studying the functionality of mitochondria, yeast cells were stained with the vital dye Rhodamine 123 (Rh123), whose incorporation depends on the mitochondrial membrane potential (MMP, ΔΨmt), and analysed via flow cytometry on FACSCalibur™ instrument (BD Biosciences, Franklin Lakes, NJ, USA). In brief, WT, *hho1∆*, *arp4* and *arp4 hho1∆* strains were grown in SCD medium, and aliquots of cells were taken at four-time points, namely the 4th, 24th, 72nd h and 9th day, washed in sterile distilled water twice and stained with 50 nM Rh123 according to Ludovico et al. [45]. After 10 min incubation with Rh123 at RT in the dark, 50,000 cells of each sample were acquired using a BD FACSCalibur™ Flow Cytometer (BD Biosciences, Franklin Lakes, NJ, USA). Flow cytometry data were analysed using the FlowJo™ v10 software (BD Biosciences, Franklin Lakes, NJ, USA).

### 2.8. Statistical Analyses

A paired-sample t-test with two-tailed distribution (Excel) was performed to analyse the significance of differences between the experimental groups. Values are reported as means ± SD from duplicate experiments, and *p* < 0.05 was considered statistically significant.

## 3. Results and Discussion

### 3.1. The Double Mutant arp4 hho1Δ Exhibits Strong “Petite” Phenotype

In previous studies, we reported the disordered chromatin structure of *S. cerevisiae arp4*, *hho1Δ*, and *arp4 hho1Δ* strains and its impact on cellular morphology, gene expression, and stress resilience well as replicative and chronological lifespan [29,33,35,46,47]. In addition to the already described phenotypes, when cell suspensions of the four studied strains were plated on solid YPD media and cultivated for three days at 30 °C, a noticeable difference in the size of the single-formed colonies depending on the strain was observed. Interestingly, all three mutant strains formed colonies with a smaller size than that of the WT (Figure 1 and Appendix A).

The order of the average colony size from the biggest to the smallest was as follows: WT > *hho1Δ ≈ arp4* > *arp4 hho1Δ* (Figure 1 and Appendix A). In addition, comparing the whisker plots data of the four strains presented in Figure 1 highlighted that the strain *arp4 hho1Δ* (harbouring the two mutations) was the most homogeneous. On the other hand, the colonies of *hho1Δ* were the most heterogeneous in size compared to the other three studied groups (Figure 1).

The *arp4 hho1Δ* mutant formed tiny colonies compared to the other three strains, with a 1.8 times smaller average diameter than the WT. A possible explanation for forming small yeast colonies when growing in glucose is the so-called “petite” phenotype. The “petite” mutation was first described by Ephrussi and co-authors [48] as a non-Mendelian mutation that causes incapability for respiration in yeast. “Petite yeasts” occur due to disturbances in mitochondrial function. In addition to the formation of petite colonies on fermentable carbon sources, these mutants are characterised by the inability of the cells to grow and form colonies on respiratory media [49], by the mtDNA deletions (rho-) or complete mtDNA depletion (rho^0^) [50,51] and mutations in nuclear genes coding for mitochondrial proteins [52]. Hereof, the colonies of the four studied strains were further velveteen replica-plated on YPGly and YPEth media containing a non-fermentable carbon source, glycerol and ethanol, respectively.

As shown in Figure 2, the WT, *hho1Δ,* and *arp4* cells propagated and formed single colonies on both types of carbon sources, fermentable and non-fermentable. At the same time, those of the double mutant *arp4 hho1Δ* could multiply only on the glucose-supplemented medium, incapable of utilising and propagating on respiratory carbon sources (Figure 2). Therefore, in addition to forming small colonies on fermentable carbon sources (glucose), the *arp4 hho1Δ* cells could not utilise two non-fermentable carbon sources, glycerol and ethanol, which is an undoubted sign of respiratory deficiency. Interestingly, one of the other mutants, *arp4,* displayed lower growth ability on ethanol when compared with WT and the *hho1* mutant (Figure 2). This also suggests a particular impairment of the mitochondrial function in the *arp4* mutant.

### 3.2. mtDNA Integrity

Mitochondria possess a multicopy genome organised in a DNA-protein complex-nucleoid—usually containing a single copy of mtDNA [53,54,55,56]. The yeast mitochondrial genome contains 35 genes coding for eight mitochondrial proteins (seven subunits of the respiratory complexes III, IV, and V and one subunit of the mitochondrial ribosome), the 9S RNA-subunit (*RPM1*) of mitochondrial ribonuclease P, two rRNAs and 24 tRNAs. In addition, several intron-encoded proteins, e.g., maturases, endonucleases, and reverse transcriptase, are encoded by the introns of *COX1*, *COB*, and *21S rRNA* genes [57,58].

In addition to the inability to utilise respiratory substrates, the “petite” phenotype in yeast is also associated with an increased frequency of mitochondrial genome deletions (rho-) or loss (rho^0^) [50,51]. The presumable “petite” status of the double mutant *arp4 hho1Δ* was manifested by its small colonies and inability to grow on YPGly and YPEth media (Figure 1, Figure 2 and Appendix A). Therefore, it was essential to determine whether the double mutant cells or any of the other three studied yeast cells had some mtDNA deficiency and could any of them be defined as rho-/rho^0^. The integrity of mtDNA in the four strains was assessed by PCR analyses examining the presence of several selected mtDNA regions. In the first set of experiments, four primers were used to amplify sequences from different regions across the *S. cerevisiae* mtDNA, namely fragments of *COX1* (primer pairs 1F/1R and 3F/3R), *21S rRNA* (primer pair 6F/6R) and *COX3* (primer pair 8F/8R) genes (Table 2, Figure 3).

Results are summarised in Figure 3. Primer pairs 1F/1R, 3F/3R and 8F/8R amplified a PCR product in WT, *hho1Δ* and *arp4* cells (amplicons I, V and XIV, respectively). These findings confirmed the presence of the corresponding mtDNA regions: 20,197–20,281, a sequence of *COX1* intron (V, 3F/3R), as well as 16,862–16,935 (I, 1F/1R) and 79,599–79,698 (XIV, 8F/8R) encompassing parts of the mitochondrial *COX1* and *COX3* coding sequences, respectively (Figure 3), in WT, *hho1Δ* and *arp4* cells. However, the primer pairs 1F/1R, 3F/3R and 8F/8R did not yield any amplified fragment in *arp4 hho1Δ*, which indicated the absence of the respective mtDNA sequences. Thus, neither the 100 bp sequence at position 79,599–79,698 of the *COX3* gene nor the fragments with coordinates 16,862–16,935 and 20,197–20,281 of *COX1* gene were present in the mitochondrial genome of the double mutant suggesting multiple short deletions, a massive deletion comprising the two examined regions of the *COX1* sequence or loss of the whole mtDNA. Unexpectedly, no amplified product was detected in all four strains when the primer pair 6F/6R was used. The results denoted a lack of mtDNA fragment with a position of 60,489–60,561 (a part of the mitochondrial *21S rRNA* gene) in all studied strains. A cross-reference was done with the above-described primer pairs in the control BY4741 (*MAT*a; *his3D1*; *leu2D0*; *met15D0*; *ura3D0*) isogenic to S288C strain, and the results were the same (data not shown).

To extend the examined regions of the mitochondrial genome, we designed an additional 13 primer pairs used in the PCR analyses to amplify upstream and downstream sequences of those assessed initially (Table 2, Figure 3). Sequences from different parts of the mtDNA, including exons and introns, were analysed to assess the integrity of the mitochondrial genome in the studied strains. Seven primer pairs, four for the *COX1* gene (1F/2R, 2F/2R, 3F/4R, 4F/4R) and three for the *21S rDNA* (6F/7R, 7F/6R, 7F/7R), yielded no amplified PCR fragments (Figure 3, dotted lines, expected amplicons III, IV, VII, VIII and XI, XII, XIII, respectively). Therefore, the results revealed the loss of the 2068 bp long mtDNA fragment (XIII) of the *21S rRNA* gene as well as several sequences of the *COX1* gene in all four strains. In PCR analyses with the other six primer pairs, three for *COX1* (2F/1R, 4F/3R, 5F/5R) and three for *COX3* (8F/9R, 9F/8R, 9F/9R), amplicons with the expected size were detected (Figure 3, solid grey lines; amplicons II, VI, IX and XV, XVI, XVII, respectively) in the WT, *hho1Δ* and *arp4* cells. Thus, no differences in the mtDNAs integrity of these three strains have been detected so far. In contrast, none of the seventeen primer pairs used for assessing the mitochondrial genome integrity gave amplification of an mtDNA fragment in the *arp4 hho1Δ* mutant cells. These findings suggest that either there is no mtDNA in the double mutant or it has undergone massive deletions compared to the reference mitochondrial genome of strain S288C. Notably, the positive control for the performed PCR reactions, the nuclear gene *ATP25*, encoding a mitochondrial inner membrane protein required for expression and assembly of the Atp9p subunit of the *S. cerevisiae* mitochondrial proton translocating ATPase [41], was successfully amplified in all examined strains, including *arp4 hho1Δ*.

The results coincided with strains’ growth ability on non-fermentable carbon sources. The double mutant *arp4 hho1Δ,* in which we detected the absence of large mtDNA regions, could not utilise glycerol and ethanol. In contrast, the single mutants and WT were all rho- (compared to the reference S288C mtDNA) but could grow on glycerol and ethanol. Variability of the mitochondrial genome size has been reported amongst different *S. cerevisiae* strains ranging from 68 kbp in strains with “short” mitochondrial genome to 86 kbp in strains with” extended” versions of the mitochondrial genome. For example, the mitochondrial genomes of *S. cerevisiae* strains D273-10B, and FY1679 (an isogenic derivative of the reference strain S288C) belong respectively to the “short” and “long” versions of the mitochondrial genome [58,59].

Interestingly, it has been revealed that the 85,779 bp “long” mitochondrial genome of FY1679 misses two fragments, each more than 1.5 kb, in comparison to the “short” D273-10B mitochondrial genome [58]. The observed strain-specific diversity in mitochondrial genome size results from differences (deletions/insertions), mainly in group I and II introns [58,60]. Unfortunately, the mtDNA of the wild-type strain DY2864 used here was not sequenced. Therefore, the mtDNA sequence of S288C was used as a reference in the present study.

Our results from the mitochondrial DNA integrity studies demonstrated the absence of approximately 2 kb fragment of the 21S rDNA and fragments in several other mtDNA regions in the WT strain DY2864 (Figure 3) compared to the reference S288C mtDNA. Similar differences in mtDNA genome size have been observed as within-species and strain-dependent variations in different strains of *S. cerevisiae* [58,59]. Thus, the deletions we found could reflect strain-specific characteristics of the DY2864 mitochondrial genome inherited by the isogenic mutant strains *hho1Δ*, *arp4* and *arp4 hho1Δ*.

Most interesting is that the introduction of the two mutations, the knockout of the *HHO1* gene and the point mutation in the *ARP4* gene, in the same yeast genome could be the reason for the observed loss of various fragments mapped at very distant locations in the mitochondrial genome (Figure 3). This loss could include multiple short deletions, several massive deletions involving the regions examined, or a complete loss of mtDNA. Therefore, the double mutant *arp4 hho1Δ* strain could be assumed as rho-. On the other hand, although *arp4* showed weaker growth on ethanol, no differences were detected between its mtDNAs and that of the wild type. We hypothesize, but so far cannot assume or rule out, a direct link between induced mutations in genes encoding chromatin proteins and the loss of large chunks of mitochondrial DNA. Further examinations are needed to reveal a possible association between chromatin mutants and mtDNA instability.

### 3.3. Mitochondrial DNA Quantification in the Course of Yeast Lifespan

Numerous theories have been postulated to explain the causes of ageing, the most extensively studied of which is the free radical, also referred to as the mitochondrial theory of ageing [61,62,63,64,65]. Increased free radical production and the concomitant accumulation of oxidatively damaged cellular macromolecules, including mtDNA, are primary drivers of ageing [66,67]. Yeast cells, as well as human ones, contain several copies of mtDNA. In humans, more copies of mtDNA per cell are found in younger individuals, and the mtDNA levels decrease with age. A reasonable exception is centenarians, who usually maintain a high mtDNA content in their mitochondria [66,68,69].

The number of mtDNA molecules depends on different factors such as growth conditions (carbon source, temperature), ploidy (haploid, diploid), ROS concentration and oxidative stress, and it has been determined that a single yeast cell can harbour between 10–50 to 200 mtDNA copies per nuclear genome [3,70,71,72]. With the progression of CLS, alterations were detected in both the number and the morphology of mitochondria, the mitochondrial network underwent extensive fragmentation, and mitochondrial filaments disappeared [11]. However, whether and how the dynamic of mitochondrial DNA copy number changes during the chronological ageing of the mutant cells with impaired chromatin structure is largely unknown.

#### 3.3.1. Relative Quantification of mtDNA by qPCR

The four strains were propagated in synthetic complete dextrose media for three days to estimate the relative copy number of mtDNA during the yeast lifespan. Aliquots of cells were taken at the 4th, 24th and 72nd hour of cultivation. The relative amount of mtDNA was assessed by qPCR, amplifying a fragment of the mitochondrial gene *COX1* as a target (primer pair 1F/1R, amplicon I) and a region of the nuclear gene *ATG18* as a reference (Table 2, Figure 3). Results presented in Figure 4 indicate that the relative number of mtDNA copies was both strain and time-dependent. As cultivation progressed, the number of mtDNAs gradually increased in WT, *hho1Δ* and *arp4* cells, which was expected as glucose is depleted over time, fermentation ceases, and respiration is derepressed. Thus, on day 3, the WT cells contained 2.8 times more mtDNA molecules than at the 4th hour (Figure 4). In *hho1Δ*, the mtDNA copy number also increased with time. However, when compared with the WT, differences were observed in the mtDNA increment. For the first 72 h, the increase of mtDNA molecules in the linker histone deficient cells was more distinct compared to the WT, 2.77-fold in WT and 4.63-fold in the *hho1Δ* mutant.

Interestingly, at the 4th hour, *arp4* cells contained almost two times more mtDNA copies than the WT, but the ratio of increase was from 1.83 at the 4th hour to only 2.4 till the 3rd day. Notably, in all three strains, the highest relative amount of mtDNA was detected at the 72nd hour. However, the *arp4* harboured lower mtDNA content of mtDNAs than the WT and *hho1Δ* strains at that time. These findings pointed out the specificity of mtDNA copy number dynamics during the chronological ageing of strains with native (WT) and impaired (*hho1Δ* and *arp4*) chromatin structure. As described above, no mtDNA was detected in the double mutant *arp4 hho1Δ* (Figure 4). Similar results were obtained when another primer pair, 8F/8R, for the amplification of fragment XIV of the *COX3* gene (Table 2) was used in the qPCR analysis (Appendix A).

#### 3.3.2. Absolute Quantification of mtDNA Copy Number

For a more precise and absolute quantification of the mtDNA copies per genome (cell), we performed selfie-dPCR, a modification of the ddPCR [43]. In three time points of the yeast culture ageing (4, 26 and 72 h), the primer pair 8F/8R was used to amplify the sequence of the *COX3* gene (XIV in Figure 3). The amplification of the single-copy genes *ATG18* and *CDC28* from nuclear DNA was used for data normalisation (Table 2). The results of selfie-dPCR for the mtDNA copy number evaluation are presented in Figure 5. Generally, the results confirmed the main conclusions based on the qPCR experiments. Again, we observed that the number of mtDNA molecules in the WT, *hho1Δ* and *arp4* strains increased with ageing and that the dynamics were strain-specific. The linker histone mutant *hho1Δ* exhibited a more than a fivefold change in the number of mtDNA molecules. In comparison, this increase was 3.5 times for WT and only 1.3 times for *arp4* (Figure 5).

Galdieri and co-authors reported that the expression of core histone genes regulates the mtDNA copy number and mitochondrial respiration. Depleting the histone H3 results in increased mtDNA copy numbers [27]. Here, we have demonstrated that the deletion of linker histone H1 leads to a similar phenotype. In contrast, the *arp4* mutant showed the highest number of mtDNA copies among the three strains at the 4th h and the lowest at the 72nd h. The highest number of mtDNA molecules/cells in all three strains was detected at the 72nd h of cultivation. Changes in the mtDNA copy number could be due to an increase in the number of mitochondria per cell and the number of mtDNA copies/mitochondrion. Again, no amplification product was yielded using the 8F/8R primer pair and total DNA extracted from the *arp4 hho1Δ* strain, irrespective of the growth time point.

*S. cerevisiae* is Crabtree-positive yeast, and when glucose concentration is high, even in the presence of oxygen, they degrade glucose mainly by fermentation [73,74]. Glucose repression affects many enzymes for metabolising other sugars and non-fermentable carbon sources. The presence of this hexose in cultural media represses the development of mitochondria and respiratory competence in *S. cerevisiae* [75]. The substrate dependency of the mtDNA copy number has been previously reported. Yeast cells from glucose-supplemented batch cultures, in which metabolism was respire-fermentative, contained 2–3 giant mitochondria per cell, while those grown on ethanol as a carbon source contained a large number (20–30) of small mitochondria [76,77,78]. Our results are in good agreement with these observations. In the first hours of cultivation in an SCD medium, yeast cells rely on glucose as a carbon source and utilise it primarily by fermentation. In young WT cultures, at the 4th hour, when glucose concentration in the media is high, cells preferably ferment it. At the same time, respiration is repressed (glucose repression on respiratory enzymes and mitochondrial development), and the number of mtDNA appears the lowest. As culturing progresses, around the 24th hour, the glucose is depleted, respiration is released from glucose repression, cells switch to aerobic respiratory growth and the copies of cellular mtDNA increase (Figure 4 and Figure 5). Moreover, our results revealed a progressive increase in the mtDNA molecules later during cultivation. This is what we observed in the WT, *hho1Δ* and *arp4* strains.

The worthy finding was that the dynamics of mtDNA copy number increase in the two mutants was different from that of the WT and particular to each one. Moreover, *arp4 hho1Δ* did not contain the numerous mtDNA fragments analysed, as no amplification was obtained with any of the primer pairs applied.

### 3.4. Rhodamine 123 Staining for Analysis of Mitochondrial Membrane Potential by Flow Cytometry

mtDNA encodes for essential subunits of the respiratory chain complexes. Therefore, the maintenance and integrity of the mitochondrial genome are crucial for oxidative phosphorylation [3]. This raises an issue about the capacity of mitochondria in the studied chromatin mutants, especially in the double mutant strain. To further investigate mitochondria’s functionality in the studied chromatin mutants during CLS, we followed the dynamics of mitochondrial potential via Rhodamine 123 staining. Flow cytometry routinely uses the lipophilic cationic fluorescent dye Rh123 to evaluate mitochondrial metabolism and activity in individual cells. The amount of incorporated Rh123 and fluorescence intensity is proportional to viable cells’ respiration-driven mitochondrial membrane potential (MMP or ΔΨmt) [79,80,81]. Therefore, cultured cells of the four yeast strains were taken appropriately, stained with Rh123 as described previously [45] and analysed in FL-1 channel (488 nm/530 nm) using the BD FACS apparatus. Cells of each sample were distributed in three populations according to the FL1-H fluorescence: FL1-H+ population comprised low fluorescence cells (cells with weak MMP); FL1-H++ (cells with high MMP) and FL1-H+++ (cells accumulated Rh123 nonspecifically). The percentage of cells in each population was calculated and presented in the charts (Figure 6 and Figure 7). As a negative control group, aliquots of WT cells were first heat-killed by incubation at 100 °C for 20 min before Rh123 staining (Figure 6A, black curves). It has been demonstrated that the heat-killed yeast cells displayed a more intense fluorescence than untreated live cells resulting from the loss of mitochondrial membrane integrity and a diffuse, non-localised distribution of Rh123 in the dead cells [45]. Hence, based on the data obtained, the authors noted that the Rh123 could be used as a probe to distinguish living from dead cells [45].

#### 3.4.1. Dynamics of the Mitochondrial Membrane Potential in Non-UV Irradiated Yeast Cells during CLS

Figure 6 represents the results of the Rh123 fluorescence as an MMP indicator of studied cells during their chronological lifespan. In WT cells, at the 4th hour, when the amount of glucose in the culture media was still high, and fermentation predominated over respiration, most of the cells (64%) emitted low Rh123 fluorescence because of the low mitochondrial activity as a result of the glucose repression. About 30% of cells belonged to the population with higher fluorescence, and only 3% displayed the non-localised distribution of Rh123 characteristics for dead cells. With the time-dependent exhaustion of glucose in the media and the transition from respire-fermentative to respiratory metabolism, the percentage of cells with low mitochondrial activity (FL1-H+) constantly diminished. At the same time, those with high MMP (FL1-H++) increased, reaching the maximum of 45% at the 72nd hour. At day 9, the number of cells in the two populations slightly decreased to 25% and 35%, respectively. Accordingly, the proportion of cells that bound Rh123 nonspecifically (dead cells) (FL1-H+++) increased along the CLS from 18% at the 72nd hour to 40% on the 9th day. In line with these results, Volejnıkova and co-authors reported that during a ten-days culture ageing, cells of the wild-type *S. cerevisiae* strain JC 482 retained their viability. At the same time, the structure and function of mitochondria changed [11].

As shown in Figure 6, apparent differences in the dynamics of Rh123 accumulation were detected in the yeast strains harbouring a mutation in *HHO1* or *ARP4* genes. The most similar to the MMP dynamics of WT was that of the *arp4* strain. However, differences have emerged between WT and mutant *arp4* cells, especially at the late time point. The population of *arp4* cells with high MMP (FL1-H++) decreased more than twice, from 37.1 ± 1.34% (for the first three time points) to 17% on the 9^th^ day (Figure 6B). In addition, at all-time points, dead *arp4* cells appeared to be more than in WT cultures, reaching 67% on the 9th day versus 40% for the WT.

Notably, the dynamics of Rh123 intake in the mitochondria of *hho1Δ* cells differed immensely from the WT and the other two mutants. In the first two time points, the 4th and 24th h, most cells were acquired as FL1-H++, i.e., cells with high MMP values, about twice as high as the WT FL1-H++ population. However, on day 9, the *hho1Δ* FL1-H++ cells dropped to 13%, a 2.7-fold decrease compared to the WT. The linker histone mutant strain showed a fast transition to FL1-H+++ fluorescence as these populations contained 51% and 85% of the cells in 72-h and nine-day cultures, respectively (Figure 6B).

For the first two time points (4th and 24th hour), the percentage of *arp4 hho1Δ* cells with FL1-H++ fluorescence resembles the results obtained for the WT and *arp4* cultures. However, of the four studied strains, *arp4 hho1Δ* cells demonstrated the fastest transition from specific to non-localised Rh123 accumulation and, concomitantly, the lowest fraction of high MMP (FL1-H++) cells at the earliest stage of CLS (Figure 6). As early as the 72nd hour, only 23% of the cells were FL1-H++. In contrast, 64% of the cells showed FL1-H+++ fluorescence characteristics for dead cells. This was 3.5, 2.1 and 1.25 times more than the corresponding population in WT, *arp4* and *hho1Δ*, respectively. That correlated with the formation of smaller colonies on YPD (Figure 1 and Appendix A) observed for the double mutant strain.

Significant perturbations in the Rh123 intake and, therefore, in the MMP dynamics were found in all three chromatin mutants during their lifespan compared to the WT strain. Some authors have found a connection between the lack of mtDNA (rho^0^), respiration deficiency and lifespan extension in rho^0^ strains. Others reported that respiration-deficient cells experience no longer but, in contrast, shorter lifespans [21,82,83]. The current and previous studies of these *S. cerevisiae* chromatin mutant strains that have been shown to age prematurely agree with the latter findings. These alterations were most pronounced in *arp4 hho1Δ* cells that could not utilise non-fermentable carbon sources, and no mtDNA was detected with the seventeen primer pairs used.

#### 3.4.2. Dynamics of Mitochondrial Potential in UVA/B Irradiated Yeast Cells during CLS

One of the main characteristics and presumed causes of ageing is the accumulation of oxidative damage inflicted by the production of free oxygen species, which increases significantly, especially with prolonged exposure to ultraviolet (UV) rays [61,63,84]. The process is known as photoageing, and it has been shown that UV-induced photoageing of skin is characterised by increased numbers of large-scale deletions in the mtDNA [84,85]. Furthermore, the mito-nuclear communication that represents the retrograde signalling pathway from mitochondria to the nucleus is also an essential part of the cell signalling network in response to internal and external stimuli [86,87,88,89,90]. For example, mitochondrial rho mutants of six yeast species manifested decreased thermotolerance than the parental wild-type strain, suggesting the mitochondrial genome’s role in adapting eukaryotic cells to critically high temperatures [91].

To address the effect of UVA/B stress on the mitochondrial activity of chromatin mutants, aliquots of WT, *hho1Δ*, *arp4* and *arp4 hho1Δ* cell cultures were irradiated with two different doses of UVA/B (460.8 mJ and 4608 mJ) and analysed for Rh123 uptake by flow cytometry (as described in Materials and methods). The results of these studies are presented in Figure 7. Compared to the untreated WT, the most significant change in the population of cells with increased MMP (FL1-H++) was detected after irradiation of 72nd-h cultured cells for 3 min (46% vs. 68%) and in the 4th h (30% vs. 61%) and 72nd h (46% vs. 65%), irradiated for 30 min cultures. Irradiation of a nine-day WT culture for 30 min also increased the portion of most strongly fluorescent (FL1-H+++) cells from 40% without UV to 56% after UV treatment. A similar response trend to UVA/B was observed for the *arp4* cultures. At both doses, 460.8 mJ and 4608 mJ, applied on the 72nd h, the WT and *arp4* cells enhanced the mitochondrial activity to cope with the applied stress. Old WT cells (9 days) were vulnerable to UV rays, only at the higher dose showing a 40% increase in the FL+++ population. The old *arp4* cultures were affected by the two doses, showing an increase in FL1-H+++ and a decrease in FL1-H++ populations.

In *hho1Δ* cells, there was no significant effect of UV on the distribution of cells population according to the Rh123 fluorescence and hence on the MMP (Figure 7). However, both 3- and 30 min irradiation affected *arp4 hho1Δ* double mutant cells similarly, causing a sharp decrease in the fraction of cells actively accumulating Rh123 at the 24th and 72nd hour. Cell fractions corresponding to those with low MMP or nonspecific accumulation of mitochondrial dye throughout the cell volume were enriched (Figure 7).

The maintenance of mitostasis, the mitochondrial homeostasis in eukaryotic cells, depends on the balance between mitofusin, mitofission, and mitophagy and is ensured by nuclear-encoded proteins [92,93,94]. In senescent cells, the healthy equilibrium is disturbed as mitofusion becomes predominant and mitochondrial turnover decreases [93,94]. During ageing and under oxidative stress, mitochondrial mass increases and senescent cells are characterised by enlarged, branched mitochondria with decreased cristae [11,95]. Our results also indicate a time-dependent decline in the mitochondrial activity upon UVA/B stress, much more noticeable in chromatin mutants than in wild-type cells.

### 3.5. Selfie-dPCR for Absolute Quantification of ATG18 and CDC28 Transcripts

Previous reports have pinpointed the role of nuclear genes in mitochondrial genome integrity, emphasising the indispensability of proper nuclear-mitochondrial communication for healthy cell development and viability [96,97,98]. Many nuclear genome-encoded proteins regulate mitochondrial morphology and function, e.g., the replication and transcription of the mitochondrial genome, which is entirely dependent on nuclear gene products [96,99,100,101]. Several lines of evidence suggest that decreased mitochondria biogenesis, mitochondrial dysfunction and the level of mtDNA content in yeast may be associated with defects in genes that regulate the cell division cycle [102,103,104]. One of these genes is *CDC28*, which regulates the meiotic and mitotic cell cycle and is associated with G2/M checkpoint cell cycle blocking [105]. Additionally, it is part of cell growth regulation, metabolism, maintenance of chromatin dynamics and morphogenesis [105]. In particular, it was found that *S. cerevisiae cdc28* mutants displayed reduced frequency of induced and spontaneous rho mutations and increased mitochondrial genome stability [102,103]. Damaged or superfluous mitochondria are removed using the autophagic machinery in the selective process of mitophagy. In yeast, as in higher eukaryotes, the nuclear-encoded protein Atg18 is a component of the Atg9•Atg2-Atg18 complex, which is crucial for autophagosomes formation [106,107]. In addition, Atg18 participates in vacuolar morphology regulation, involving binding by phosphatidylinositol 3,5-bisphosphate [108].

In chromatin mutants, the compromised chromatin organisation and epigenetic regulation could lead to altered expression of genes involved in mitochondrial biogenesis, dynamics and function. Selfie-digital PCR (selfie-dPCR) was designed to precisely quantify the number of transcripts per genome by the gene of choice [43,44]. Here we applied the selfie-dPCR technique to assess precisely *ATG18* and *CDC28* genes’ transcription. *ATG18* and *CDC28* mRNAs were estimated in three-time points, the 4th, 24th and 72nd h, in the WT, *hho1Δ, arp4,* and *arp4 hho1Δ* strains and results are presented in Table 3. In the WT strain, the number of transcripts per genome of *ATG18* and *CDC28* genes steadily decreased with the cultivation time (Table 3), while increased mtDNA copies were detected (Figure 4, Figure 5 and Appendix A). Similar to WT, the expression of both genes in the *hho1Δ* and *arp4* single mutants exhibited a time-dependent reduction. However, in the three chromatin mutants, the number of transcripts of the two genes was lower than that of the WT strain at all the time points studied. Furthermore, the extremely low basal (4th h) *ATG18* and *CDC28* transcripts were detected in the *arp4 hho1Δ* mutant, which exhibited an apparent petite phenotype and large mtDNA deletions (rho-). These results follow the results obtained with RT-qPCR (Appendix A).

Atg18 was revealed to play an essential role in maintaining the structural integrity of mitochondria in fruit fly heart cells [107] and regulating mtDNA content in *C. elegans* [109]. Furthermore, Hibshman and co-authors reported that the autophagy mutant *atg-18/Atg18* retained more mtDNA copies than WT worms. Similarly, in the yeast WT, *hho1Δ*, and *arp4*, we observed a time-dependent decrease in *ATG18* and *CDC28* expression (Table 3) and an increase in mtDNA copy number (Figure 4 and Figure 5).

The crosstalk between the nucleus and mitochondria is crucial since in the yeast *S. cerevisiae,* all but sixteen (thirteen in *Homo sapiens*) mitochondrial proteins are encoded by the nuclear genome [58,110]. Many nuclear genes are involved in the replication, transcription, integrity, and segregation of mitochondrial DNA. In humans, mutations in genes that lead to mitochondrial genome instability are associated with mitochondrial diseases characterised by mtDNA multiple deletions or depletion [3,111,112,113]. For example, in primary mouse embryonic fibroblasts, p53 controlled mtDNA copy number, mitochondrial genome stability, mito-checkpoint pathway and mitochondrial biogenesis [101]. Significantly, most mtDNA genes are conserved between yeast and humans, making yeast a preferable model organism for studying mitochondrial diseases [3]. On the other hand, in yeast rho^0^ cells, the lack of mtDNA affects respiration and mitochondrial-nuclear communication [21].

Differences in the dynamics of the mtDNA copy number and MMP amongst the four studied yeast strains detected in the current study may result from the particular chromatin structure and altered regulation of the gene expression as a consequence, as detected for *ATG18* and *CDC28* genes. This could disturb the signalling from the nucleus to the mitochondria and the retrograde mitochondria-to-nucleus communication. Moreover, it has been shown only recently that the communication between mitochondria and the nucleus coordinates the epigenetic regulation of cellular response to external cues [90]. For example, chromatin structure has been found to influence mtDNA content as improper chromatin assembly or reduction in histone H3 and H4 levels in *S. cerevisiae* has been shown to promote mtDNA copy number increment [27]. On the one hand, during respiratory growth, to remodel the chromatin, ATP-dependent Chromatin Remodelling Complexes (CRCs; Arp4 is a subunit of three CRCs in yeast) rely on the ATP produced by mitochondria.

The chromatin structure is one of the epigenetic factors regulating gene expression. The three mutant strains, *hho1Δ*, *arp4* and *arp4 hho1Δ,* display disordered chromatin organisation and strain-specific chronological and replicative ageing [33,34,35,36,47]. Therefore, we argue that, most probably, the disordered chromatin organisation in the studied mutants leads to altered expression of many genes, affecting the communication between the nucleus and mitochondria, which results in changes in mtDNA copy numbers and the premature mitochondrial dysfunction that we observed.

## 4. Conclusions

Mitochondria are multifunctional cellular organelles with an endosymbiotic origin, essential for eukaryotic cells. Mitochondria are the cell’s powerhouse, generating most of the cellular ATP. Moreover, the process of ATP synthesis is coupled with the production of reactive metabolic intermediates and reactive oxygen species that can serve as signalling molecules for the modulation of cellular metabolism and critical regulators of cell fate [64,111,114]. Thus, mitochondria play a central role in the stress response, cell longevity, ageing and death (mitochondria-mediated apoptosis and necrosis) [87,88,89,98,111].

Amongst the recently updated eleven cellular markers of ageing postulated so far, genomic instability and epigenetic alterations have been described as primary hallmarks (causes of damage) and mitochondrial dysfunction as antagonistic hallmarks (responses to damage) [115]. Genomic instability and epigenetic alterations are accompanied by specific metabolic changes that affect or depend on mitochondria and mito-nuclear communication [116]. The vital role of the functional state of mitochondria and mitochondrial metabolomics in the yeast lifespan has been reported [11,21,117]. Chronological ageing is a complex, multifactorial process in which communication and interaction among all cellular compartments and systems are essential. That includes communications between mitochondria and other cellular compartments and structures such as the nucleus, the vacuoles, the peroxisomes, the plasma membrane, the endoplasmic reticulum, and the cytosol during the entire yeast lifespan [118,119]. In addition, the epigenetic communication between nuclear and mitochondrial genomes, which occurs at multiple levels, provides coordinated gene expression in response to internal and external signals [90]. *S. cerevisiae* strains with compromised chromatin structure showed premature ageing phenotypes and mitochondrial perturbations. Together, the results from the study on mitochondrial DNA integrity revealed the loss of a 2068 mtDNA fragment of the *21S rDNA* and several sequences of the *COX1* gene in all four strains compared to the reference S288C mitochondrial genome. In addition to forming small colonies and the inability to propagate on a non-fermentable carbon source, no amplification was achieved on the double mutant *arp4 hho1Δ* template using the seventeen primer pairs. The examined chromatin mutants *hho1Δ*, *arp4* and *arp4 hho1Δ* showed altered MMP dynamics during CLS and aggravated disturbances after UV irradiation. Combining the two mutations (*hho1Δ* and *arp4*) in one genome (*arp4 hho1Δ* strain) resulted in mtDNA depletion, respiratory deficiency and mitochondrial dysfunction, suggesting the rho- status of *arp4 hho1Δ* cells. These results highlighted that chromatin structure is essential for communicating between mitochondria and the nucleus. The number of mtDNA molecules and nuclear-encoded *ATG18* and *CDC28* transcripts were affected in chromatin mutants. However, many other nuclear-encoded factors, beyond Atg18p and Cdc28p, regulate copy number, the integrity of mitochondrial genomes, and mitochondrial function. The disclosure of these complex regulatory mechanisms deserves to be studied in detail.

## Figures and Tables

**Figure 1 jof-09-00329-f001:**
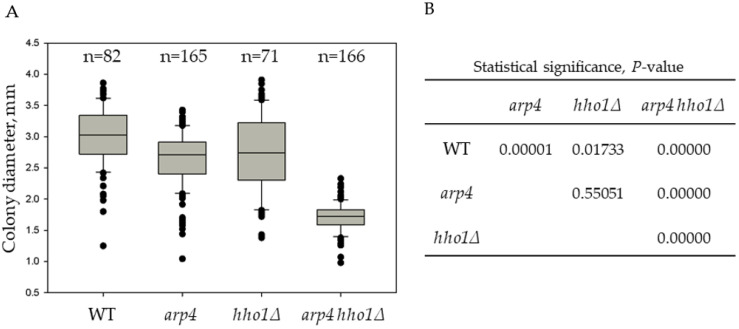
Heterogeneity of the yeast colonies from the studied strains based on their diameter distribution. (**A**). Box and whisker plots of yeast colonies’ size. Cell suspensions from overnight cultures were appropriately diluted, plated on YPD plates and incubated for three days at 28 °C to allow the formation of single colonies. The diameter of all colonies from the duplicated experiment was measured to determine the average colony diameter (d_av_, mm). The number of all analyzed colonies is indicated by n. (**B**). Results from the analysis for statistical significance of differences, *p*-value.

**Figure 2 jof-09-00329-f002:**
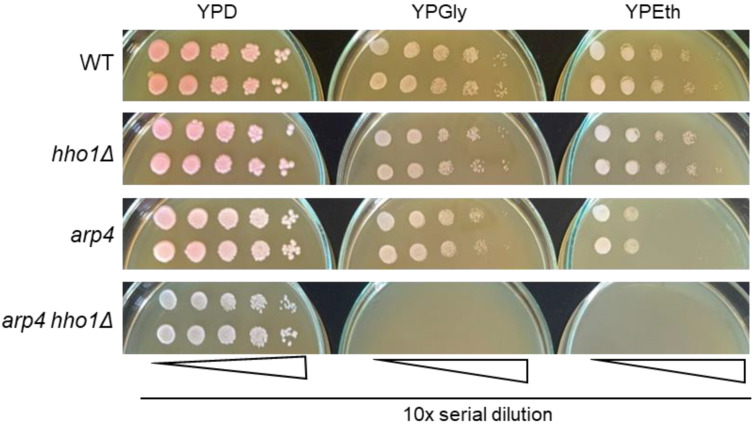
Replica plates of serial dilutions of single colony suspensions of each strain on YPD, YPGly and YPEth. Cell growth was documented after three days of incubation at 28 °C. Representative images are displayed from three repetitions of the experiment.

**Figure 3 jof-09-00329-f003:**
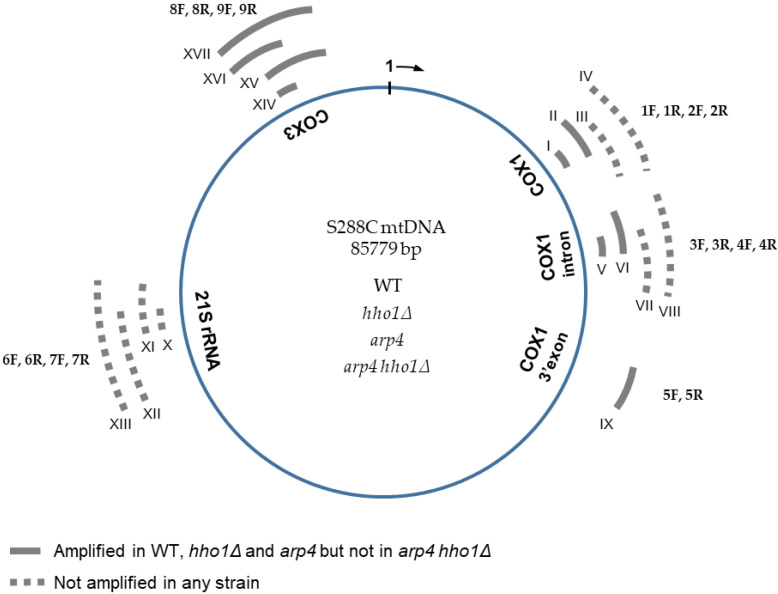
Schematic representation of *S. cerevisiae* mitochondrial DNA with five examined gene regions. The sequence of the S288C mitochondrial genome was used as a reference. Nucleotide sequences of the 18 oligonucleotides (nine forward, designated from 1F to 9F, and nine reverse, from 1R to 9R) used for the mtDNA integrity analyses are described in Table 2. Roman numerals from I to XVII indicate the expected PCR products (corresponding to the 17 primer pairs used). The position and the size of the expected amplicons are also given in Table 2. Sequences amplified with each primer pair 1F/1R and 8F/8R, used for relative or absolute (ddPCR) qPCR estimation of the mtDNA copy number in the four studied strains WT, *hho1Δ*, *arp4* and *arp4 hho1Δ,* were I and XIV, respectively. Two nuclear genes, *ATG18* (coding the phosphoinositide binding protein, Atg18p) and *ATP25* (coding the mitochondrial inner membrane protein, Atp25p), were used as controls in the PCR reactions.

**Figure 4 jof-09-00329-f004:**
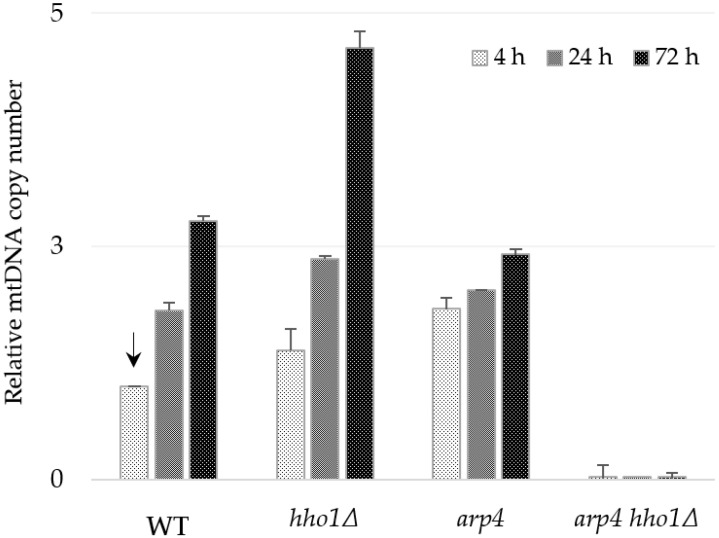
Relative mtDNA copy number in the WT, *hho1Δ*, *arp4* and *arp4 hho1Δ* cells measured by real-time PCR with the primer pair 1F/1R of the mt*COX1* gene. The single copy nuclear gene *ATG18* was used as a reference gene for normalization, and the sample WT 4 h was applied to calibrate the results (an arrow indicates calibrator). The primer pair 1F/1R is complementary to the *COX1* gene exon sequence, and the amplified fragment is designated as “I” in Figure 3 and Table 2.

**Figure 5 jof-09-00329-f005:**
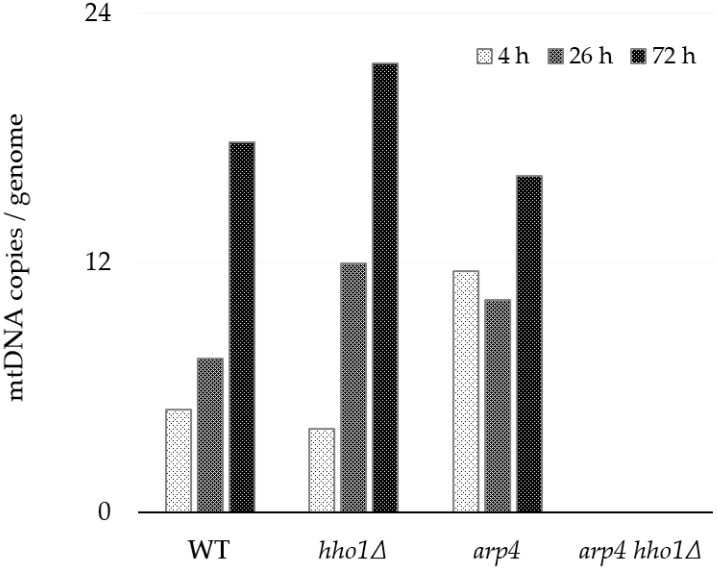
Absolute quantitation of the mtDNA number in WT, *hho1Δ, arp4* and *arp4 hho1Δ* per genome assessed by droplet digital PCR (ddPCR) using primer pair 8F/8R for mt*COX3* gene. In Figure 3 and Table 2, the corresponding amplified PCR product is denoted as XIV. The average copy numbers of the single-copy *ATG18* and *CDC28* genes were used to estimate nuclear genomes. The strains are haploids; each gene has one allele per nuclear genome.

**Figure 6 jof-09-00329-f006:**
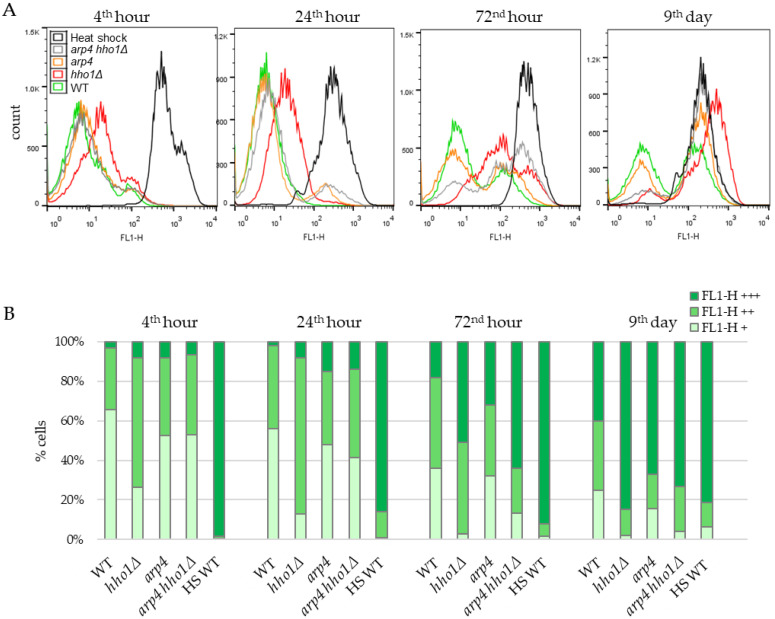
Dynamics of the mitochondrial membrane potential in the non-UV irradiated yeast cells during CLS assessed by FACS after staining with Rhodamine 123. (**A**). Histograms. For estimation, cells of each strain WT, *hho1Δ, arp4* and *arp4 hho1Δ* were separated in three sets according to the FL1-H fluorescence intensity: low, basal (FL1-H+, 0–10) characteristic for fermenting cells; high (FL1-H++, 10–100) for respiring cells and over 100 (FL1-H+++) for non-localized distribution of Rh123 in nonviable cells. In addition, heat-shocked WT cells (HS WT) were used as a negative control group. (**B**). Quantification of the number of cells in each set according to the fluorescence intensity.

**Figure 7 jof-09-00329-f007:**
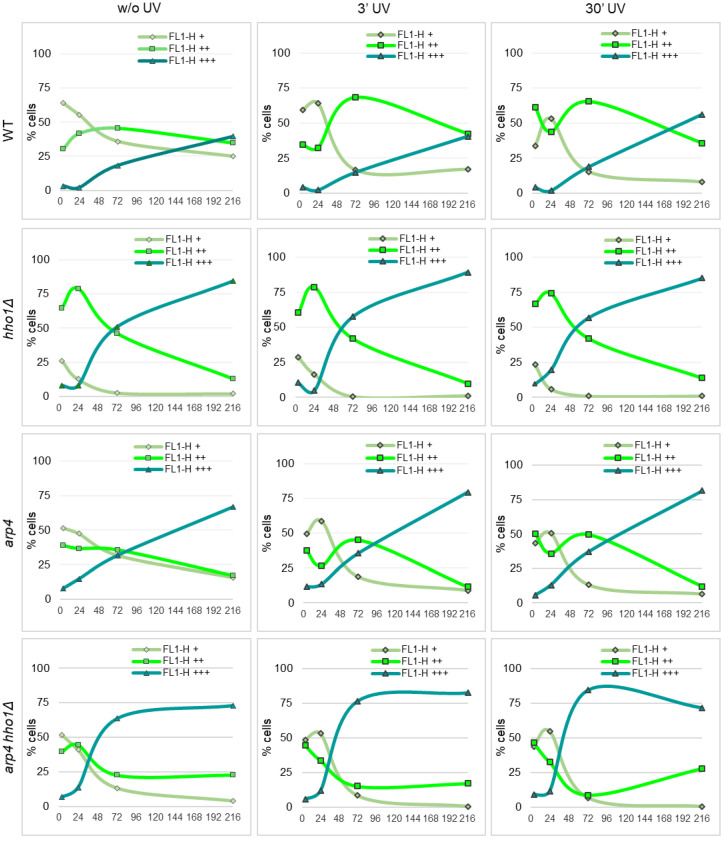
Fluorescence of Rh123 accumulation in the yeast cells after irradiation with UVA/B acquired by flow cytometry at FL1-H. Trend lines denote the proportion of cells in FL1-H+ (low MMP), FL1-H++ (high MMP) and FL1-H+++ (non-localized accumulation of Rh123 in dead cells) fractions in the course of the CLS of non-UV and UV irradiated (3 min and 30 min) cell populations of WT, *hho1Δ*, *arp4* and *arp4 hho1Δ*.

**Table 1 jof-09-00329-t001:** Yeast *S. cerevisiae* strains.

Strain		Genotype	Ref.
Abbreviation	Name		
WT	DY2864	*MAT*a *his4-912δ-ADE2 his4-912δ lys2-128δ can1 trp1 ura3 ACT3*	[39]
*hho1Δ*	DY2864 hho1Δ	*MAT*a *his4-912δ-ADE2 his4-912δ lys2-128δ can1 trp1 ura3 ACT3 YPL127C::KLURA3*	[29,36]
*arp4*	DY4285	*MAT*a *his4-912δ-ADE2 lys2-128δ can1 leu2 trp1 ura3 act3-ts26*	[39]
*arp4 hho1Δ*	DY4285 hho1Δ	*MAT*a *his4-912δ-ADE2 lys2-128δ can1 leu2 trp1 ura3 act3-ts26 YPL127C::KLURA3*	[29,36]

**Table 2 jof-09-00329-t002:** Primer pairs used in the PCR analyses.

Primers		Gene	Amplicon №	* Coordinates	** bp
Mitochondrial Gene				mtDNA	
1F	GGAGGGCTGTACGAGTTCAA	*COX1* exon	I	16,862–16,935	74
1R	CGTTATCCCCAGGGTTTCCC				
2F	ATGATTTTCTGTGCGCCGTT	AI2/*COX1*	II	16,461–16,935	475
1R	CGTTATCCCCAGGGTTTCCC	*COX1* exon			
1F	GGAGGGCTGTACGAGTTCAA	*COX1* exon	III	16,862–17,696	835
2R	ACTGACAACACTACCTTGAGGA	AI2/*COX1*			
2F	ATGATTTTCTGTGCGCCGTT	AI2/*COX1*	IV	16,461–17,696	1236
2R	ACTGACAACACTACCTTGAGGA				
3F	CCTCGCGGGGTATGGTAAAT	*COX1* intron	V	20,197–20,281	85
3R	GCATGGGGGTGGGGAAATTA				
4F	TACTTTCTTCCCCTCCGAATCC	*COX1* intron	VI	20,009–20,281	273
3R	GCATGGGGGTGGGGAAATTA				
3F	CCTCGCGGGGTATGGTAAAT	*COX1* intron	VII	20,197–20,955	759
4R	AGATTGGGTCACCACCTCC	*COX1* exon			
4F	TACTTTCTTCCCCTCCGAATCC	*COX1* intron	VIII	20,009–20,955	947
4R	AGATTGGGTCACCACCTCC	*COX1* exon			
5F	TGGGTGCTATTTTCTCTTTATTTGC	*COX1*_3′-ex	IX	26,244–26,673	430
5R	GAGTGTACAGCTGGTGGAGA				
6F	CTCTCGGTGGGGGTTCACAC	*21S rRNA*	X	60,489–60,561	73
6R	GACCCGAAAGGGAACCGGAA				
6F	CTCTCGGTGGGGGTTCACAC	*21S rRNA*	XI	60,489–61,921	1433
7R	ATCGAGGTGGCAAACATAGC	*21S*/SCEI			
7F	GCCTATAATTGAGGTCCCGC	*21S*/SCEI	XII	59,854–60,561	708
6R	GACCCGAAAGGGAACCGGAA	*21S rRNA*			
7F	GCCTATAATTGAGGTCCCGC	*21S*/SCEI	XIII	59,854–61,921	2068
7R	ATCGAGGTGGCAAACATAGC				
8F	TGAAGCTGTACAACCTACCGA	*COX3*	XIV	79,599–79,698	100
8R	ACCTGCGATTAAGGCATGATGA				
8F	TGAAGCTGTACAACCTACCGA	*COX3*	XV	79,599–80,017	419
9R	CTCCTCATCAGTAGAAGACTACG				
9F	AGAAGTAGACATCAACAACATCCA	*COX3*	XVI	79,228–79,698	471
8R	ACCTGCGATTAAGGCATGATGA				
9F	AGAAGTAGACATCAACAACATCCA	*COX3*	XVII	79,228–80,017	790
9R	CTCCTCATCAGTAGAAGACTACG				
**Nuclear mt gene**				**Gene name**	
ATP25_For	CCACCACACGATGAACAAAGA	*ATP25*		*YMR098C*	1041
ATP25_Rev	TGACTAGAATGCTGCGTTTTCA				
**Nuclear gene**				**Gene name**	
ATG18_For	TTCCCGTTGAAACCAATTCCCA	*ATG18*		*YFR021W*	98
ATG18_Rev	GCCAGTTTCGAAGAGTTCCGGAT				
CDC28_For	AGGAAACCAATCTTCAGTGGCGA	*CDC28*		*YBR160W*	91
CDC28_Rev	CTGGCCATATAGCTTCATTCGGC				

* Coordinates of expected amplicon according to the reference S288C mitochondrial genome, bp. ** Expected amplicon size, bp.

**Table 3 jof-09-00329-t003:** Quantitation of *ATG18* and *CDC28* transcripts number per genome by droplet digital PCR (ddPCR). MEAN values are the average of two experiments ± SD.

	*ATG18*			*CDC28*		
	4 h	24 h	72 h	4 h	24 h	72 h
WT	21.4 ± 1.6	3.5 ± 0.0	1.5 ± 0.5	67.6 ± 0.0	8.5 ± 1.9	1.6 ± 0.2
*hho1* *Δ*	3.7 ± 1.0	1.6 ± 0.3	0.3 ± 0.03	10.1 ± 1.4	1.6 ± 0.0	0.8 ± 0.01
*arp4*	13.0 ± 2.4	7.4 ± 3.2	0.7 ± 0.0	40.0 ± 2.8	3.1 ± 0.07	0.7 ± 0.0
*arp4 hho1* *Δ*	0.7 ± 0.1	0.5 ± 0.04	1.2 ± 0.3	1.5 ± 0.4	1.3 ± 0.1	0.8 ± 0.4

## Data Availability

Not applicable.

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
