# Peer review of "Yeast Chromatin Mutants Reveal Altered mtDNA Copy Number and Impaired Mitochondrial Membrane Potential"

_jof, 2023, doi:10.3390/jof9030329_

Round 1

Reviewer 1 Report

In this manuscript, Staneva et al. study mitochondrial dynamics of ageing in budding yeast chromatin mutants. In particular, the authors investigated arp4, hho1 and double mutants. Remarkably, the latter displays strong petite phenotype and depletion of mtDNA. Overall, the work seems to be well performed and convincing. However, some controls are missing in some experiments. In addition, a couple of assays need to be repeated to obtain statistical significance. Please see my comments below.

_Fig. 4 and 5: Experiments need to be repeated at least once to obtain error bars. Otherwise, few conclusions can be drawn from these experiments, except that double mutant is lacking mtDNA.

On the other hand, it is unclear what the starting point is. Are cells already aged at point 0? Ideally, a arp4/hho1 heterozygous diploid would generate the four haploid genotypes of interest and cultures can be started from that point.

Table 3: same as above. Please repeat the experiment to obtain error bars.

_Fig. 3: A control for 21SrRNA primers needs to be included (e.g. DNA from S288C strain). In addition, the authors can design oligos outside the deleted region and obtain junction PCR products. Ideally, Southern blots would tell more details about rearrangements and copy number.

_ Double mutant seems to be rho-. This should be confirmed by DAPI staining and/or Southern.

_ Fig. 1B: Please state N of colonies analyzed.

_ Please remove pilcrow symbols (¶) from figures

_ Nomenclature consistency: Italics are not always used for genotypes.

Author Response

Dear Reviewer,

Thank you very much indeed for the valuable remarks on our manuscript. We have followed them and revised the paper accordingly.

Please, find below our detailed replies to all remarks:

  1. 4 and 5: Experiments need to be repeated at least once to obtain error bars. Otherwise, few conclusions can be drawn from these experiments, except that double mutant is lacking mtDNA.

The data presented in Figure 4 for the relative quantification of mtDNA copy number using 1F/1R (COX1) primer pair are the mean of two independent experiments with error bars included. Values are expressed as MEAN ± SD. In addition, in Supplementary Materials, Figure S2 presents the results of relative qPCR with 8F/8R (COX3) primer pair. The relative mtDNA levels estimated using the two primer pairs (1F/1R and 8F/8R) are comparable and are from two repetitions, as already denoted.

So, the measurement of absolute mtDNA copy number by ddPCR (Figure 5) was performed not to determine the exact number but to confirm the observed trends in different strains already presented in Figure 4. Therefore, we agree that it would have been better for the digital droplet PCR to be repeated. Still, unfortunately, this is impossible for now as these experiments are costly, and the team of Dr Podlesniy has provided the machinery to do.

 On the other hand, it is unclear what the starting point is. Are cells already aged at point 0? Ideally, a arp4/hho1 heterozygous diploid would generate the four haploid genotypes of interest and cultures can be started from that point.

This is a valuable remark. Thank you! The cells are not aged at point 0, but in the timeline of their chronological lifespan, the chromatin mutants age faster, with the double mutant being the most affected. We prove these results in our previous papers, cited in the text. For the experiments that study the change in mitochondria and gene expression in the timeline of the ageing process, the time points were the 4th, 24th and 72nd hours of cultivation. These data points are denoted in the text. Time points 4th and 24th hours are points where cells are logarithmically growing.

Table 3: same as above. Please repeat the experiment to obtain error bars.

The needed data from the repeated experiment are added. We further added Suppl. Table 1 presents the relative quantitation of ATG18 and CDC28 transcripts levels by RT-qPCR as MEAN values of two experiments ± SD.

  1. 3: A control for 21SrRNA primers needs to be included (e.g. DNA from S288C strain). In addition, the authors can design oligos outside the deleted region and obtain junction PCR products. Ideally, Southern blots would tell more details about rearrangements and copy number.

We have performed additional PCRs with the reference strain S288C. The results are shown in the Table. Total DNA of the wild-type S288C isogenic yeast strain BY4741 (MATa; his3D1; leu2D0; met15D0; ura3D0) was isolated after 72 hours of cultivation in YPD medium at 30°C. The results of the additional analyses followed the previously obtained results. With some primer pairs, incl. those targeting 21S rDNA, no amplification was achieved in any of the strains examined. With other primer pairs, e.g. 9F/9P, a fragment was amplified in the wild-type strains but not in the double mutant, while with the positive control, the ATP25 gene, amplification was present in all strains.

The mitochondrial genome size has been reported to vary amongst different S. cerevisiae strains ranging from 68 kbp in strains with ”short” mitochondrial genome to 86 kbp in strains with ” extended” versions of the mitochondrial genome [81,82, Foury et al., 1998; Tzagoloff & Myers, 1986]. The observed strain-specific diversity in mitochondrial genome size results from differences (deletions/insertions), mainly in group I and II introns [81,83, Foury et al., 1998; Dujon, 1989], of which 21S rDNA is abundant. For example, it has been revealed that a “long” mitochondrial genome misses fragments, of more than 1.5 kb, in comparison to a “short” mitochondrial genome [81, Foury et al., 1998]. Therefore, additional detailed analyses must be performed to determine what version of the mtDNA the analysed strains possess.

  1. Foury, F.; Roganti, T.; Lecrenier, N.; Purnelle, B. The complete sequence of the mitochondrial genome of Saccharomyces cerevisiae. FEBS Lett 1998, 440, 325-331, doi:10.1016/s0014-5793(98)01467-7.
  2. Tzagoloff, A.; Myers, A.M. Genetics of mitochondrial biogenesis. Annu Rev Biochem 1986, 55, 249-285, doi:10.1146/annurev.bi.55.070186.001341.
  3. Dujon, B. Group I introns as mobile genetic elements: facts and mechanistic speculations--a review. Gene 1989, 82, 91-114, doi:10.1016/0378-1119(89)90034-6.

We did not include these data in the text as it is lengthy and complicated. However, these results are summarised in one sentence in Line 300: “A cross-reference was done with the above-described primer pairs in control S288C strain, and the results were the same (data not shown).”

  1. Double mutant seems to be rho-. This should be confirmed by DAPI staining and/or Southern.

Summarising all results obtained so far, the double mutant seems to be rho- not rho0: with small colonies, inability to utilise respiratory carbon sources, and mtDNA deletions. However, FACS analysis with Rhodamine 123 detected active MMP suggesting still functional mitochondria, although at a lower level. In this case, it would be difficult to discriminate between rho+ and rho- by DAPI staining.

In that particular case of rho-, considering the significant number of primers we used without any amplification, sequencing of mtDNA would give the definitive result, and we plan to perform it in our future research.   

  1. 1B: Please state N of colonies analysed.

                        Done.

The total number of colonies analysed for each strain was specified in Figure 1:

NWT = 82;        Narp4 = 165;       Nhho1Δ = 71;        Narp4 hho1Δ = 166

  1. Please remove pilcrow symbols (¶) from figures – Done.

The pilcrow symbols (¶) were removed from the figures

  1. Nomenclature consistency: Italics are not always used for genotypes.

The text was carefully checked, and all noticed genotypes were italicised.

Author Response

Dear reviewer,

Thank you very much indeed for the valuable remarks!

We have addressed them carefully and now present the revised version of our manuscript.

  General Comments

  1. Text rather describes the observations and does not add much to better understanding of aging process.

The text was reorganized, and some paragraphs were rewritten – please refer to the track changes in the revised manuscript. The stressing point is put on the nuclear-mitochondrial interactions in the studied strains, the mtDNA copy number and its dynamics in the ageing process. We highlight the role of mitochondria in ageing in this manuscript as the role of chromatin has been studied in detail by other authors and ourselves, and the references are provided in the revised text.

  1. Text is not well organized. Some chapters do not refer to the figures. Some figures show reproductory results.

The text was reorganized so that figures showing repeated results were removed from the body text, and all chapters refer to the figures now.

  1. In Introduction the information about the function of Arp4 and Hho1 is completely lacking.

Information about the function of actin-related protein 4, Arp4p and the linker histone H1, Hho1p are included in the Introduction. Please see Line 82-93 “In our previous studies, we have proven that the physical interaction between two crucial chromatin components – the yeast linker histone Hho1p, responsible for nucleosome stabilization and higher order chromatin structure maintenance [29-31], and the actin-related protein Arp4p – a subunit of Ino80, SWR1 and NuA4 [32], was of utmost importance for both yeast replicative and chronological ageing. The studied chromatin mutants were arp4 (with a point mutation in the ARP4 gene, coding for actin-related protein 4 - Arp4p), hho1Δ (lacking the HHO1 gene, coding for the linker histone H1), and the double mutant arp4 hho1Δ cells with the two mutations. We have shown that the disruption of the interaction between the linker histone and Arp4p affected the organization of chromatin structure, cellular morphology, and how cells responded to stress. Moreover, the double mutant cells that experienced premature ageing phenotypes had a reduced chronological and replicative lifespan and lowered replicative potential [33-37].”

  1. We don’t know why respective arp4 and hho1 mutants and particular double mutant are so interesting to study.

Our previous research reported the impaired chromatin organization in arp4, hho1Δ and arp4 hho1Δ mutants and the compromised chronological and replicative lifespan, especially in the double mutant. To go into greater detail about the mechanisms of the observed premature ageing phenotype, we focused on the mitochondrial function as one of the actors of the mito-nuclear communication axis. Moreover, we observed the smaller size of arp4 hho1Δ colonies in our previous studies and supposed that these petite colonies may result from cellular mitochondrial dysfunction. It is already described in the text. See above and in many other places in the revised version.

  1. Too many references (141) are listed for such experimental article.

            We have reduced the number of references and shortened the text, mainly the Introduction part.

  1. New references are lacking; 2023-1, 2022-0, 2021-3, 2020-4. Nothing was published on yeast aging in 2022? Looks like the text was written in 2021. Should be updated, better have 25% from last 5 years.

We have updated some of the references. However, we attempt to correct and cite the original article regardless of how “old” it is.

Title

Title is not informative.

The title was changed: “Yeast chromatin mutants reveal altered mtDNA copy number and impaired mitochondrial membrane potential”

Abstract

No conclusion about possible mechanisms of accelerated aging in mutants studied is present on the end. Only observations.

We have provided a conclusion. Please see  “The results suggest that in the studied chromatin mutants, hho1Δ, arp4 and arp4 hho1Δ, the nucleus-mitochondria communication is disrupted, leading to impaired mitochondrial function, which is a prerequisite for the premature ageing phenotype in these mutants, especially in the double mutant.”

Introduction

Information about mutants studied (hho1, arp4), function of proteins encoded in mutant genes (Hho1, Arp4) is completely lacking.

А paragraph with the required information is inserted in Introduction:

“In our previous studies, we have proven that the physical interaction between two crucial chromatin components – the yeast linker histone Hho1p, responsible for nucleosome stabilization and higher order chromatin structure maintenance [29-31], and the actin-related protein Arp4p – a subunit of Ino80, SWR1 and NuA4 [32], was of utmost importance for both yeast replicative and chronological ageing. The studied chromatin mutants were arp4 (with a point mutation in the ARP4 gene, coding for actin-related protein 4 - Arp4p), hho1Δ (lacking the HHO1 gene, coding for the linker histone H1), and the double mutant arp4 hho1Δ cells with the two mutations. We have shown that the disruption of the interaction between the linker histone and Arp4p affected the organization of chromatin structure, cellular morphology, and how cells responded to stress. Moreover, the double mutant cells that experienced premature ageing phenotypes had a reduced chronological and replicative lifespan and lowered replicative potential [33-37].

The reason why these particular mutants were chosen for analysis is lacking.

Our previous research reported the impaired chromatin organization, compromised chronological and replicative lifespan, and premature ageing phenotype in arp4, hho1Δ and arp4 hho1Δ mutants. These phenotypes were most strongly manifested in the double mutant cells. During these analyses, we noticed the smaller size of arp4 hho1Δ colonies than that of the wild type and single mutant strains. This observation raised the hypothesis for respiratory deficiency of arp4 hho1Δ cells. It is well known that mitochondrial dysfunction is one of the causes and hallmarks of ageing. The current study aims to investigate mitochondrial fitness/sickness using the three chromatin mutants as a model system and to shed some light on mito-nuclear axis maintenance and the processes leading to accelerated ageing.

In the present article itself, it is not correct to state we examine chronological ageing as the time points analyzed are up to 72 h (except for MMP), so when conclusions involve CLS, we refer to our previous

In our previous studies, we observed the smaller size of arp4 hho1Δ colonies and supposed that these petite colonies may result from cellular mitochondrial dysfunction. To go into greater detail about the mechanisms of the observed premature ageing phenotype, we focused on the mitochondrial function as one of the actors of the mito-nuclear communication axis and one of the hallmarks of ageing.

All these observations are briefly stated in the revised version of the manuscript.

Figure 1

A.1. White colonies are not well visible on white background. The background should be black , as is in Figure 2A.

To clarify and avoid presenting figures showing duplicated results in the main text, Figure 1A is now included as Figure S1 in the Supplementary materials.

A.2.  Writing on plates should be removed before taking image  

Panel A of Figure 1 is now presented as Figure S1. The representative pictures do not contain handwritten inscriptions.

A.2. Lines should clearly indicate which numbers were compared in statistical analysis. More lines is needed. Vertical dash on the end of each line should be added. Otherwise is not clear what was compared.

The mean values from two experiments were used for statistical analysis for each strain. In each experiment, at least 30 colonies/strain were measured. A vertical dash on the left end of each line was added to indicate the strain, which was compared to another studied strain for the significance of colony size differences. For example, WT was compared to any of the chromatin mutants, and the P-values are given below the respective mutant image. arp4 was compared to the other two chromatin mutants, hho1Δ and arp4 hho1Δ.

A.4. It would be better to show statistical analysis in the graph B.

A table with P-values for the statistical significance of differences is introduced in Figure 1.

  1. How many colonies were analyzed in each experiment? How many times experiment was repeated? State it clearly. Looks like 10 colonies of each strain were measured?. One experiment? Definitely not sufficient.

Number of colonies measured for determination of average colony size of WT, arp4, hho1Δ and arp4 hho1Δ strains

WT

arp4

hho1Δ

arp4 hho1Δ

colonies, n=

Average diameter

colonies, n=

Average diameter

colonies, n=

Average diameter

colonies, n=

Average diameter

Experiment 1

44

3.09 ± 0.24

62

2.69 ± 0.18

29

2.756 ± 0.26

73

1.71 ± 0.10

Experiment 2

38

2.89 ± 0.23

103

2.62 ± 0.24

42

2.737 ± 0.34

93

1.69 ± 0.11

Average 1+2

3.00 ± 0.08

2.66 ± 0.03

2.75 ± 0.008

1.7 ± 0.011

Experiment 1+2 all colonies

82

3.0 ± 0.24

165

2.65 ± 0.22 

71

2.75 ± 0.31

166

1.7 ± 0.11

Figure 2

Part A and B are repetitions of the same result. B is sufficient. - Done

Panel A is removed. Figure 2 consists of panel B alone.

Figure 3

It is not clear from the legend what 4 circles correspond to. Better remove thin lines, are misleading. - Done

Thin lines of the 4 circles were removed.

Figure 4

Not explained in the legend what means the abbreviation on the axis. - Done

The abbreviation on the Y-axis was removed, and the Y-axis title was inserted, “Relative mtDNA copy number.”

Figure 6

No description of colors in A, also not in the legend. L561 cells of each strain. Genes in italics in B and space between genes in genotype should be added. - Done

Figure 6, panels A and B were revised according to the reviewer’s recommendations: 

Description of colours was included in Figure 6A; In Figure 6B, genes were italicized, and space between genes in genotype was added; “cells of each group” was corrected to “cells of each strain”.

Specific comments

P3, L140, 1%, no space – corrected: 1 % to 1%

P4, L153, abbreviation ON is not explained – corrected: “ON” is replaced with “overnight”

P12, L395, mutations in genes encoding chromatin proteins – corrected: “mutations in chromatin proteins” is replaced with “mutations in genes encoding chromatin proteins”

P12 chapter 3.3, There is no reference to the figure or table in this part. Should no constitute a chapter. Should be connected to the next chapter? – changed:

Chapters 3.4. and 3.5. are now connected to chapter 3.3. as subsections 3.3.1. and 3.3.2., respectively.

 P14, L466 histon H1 for the first time, too late. More information needed. - Done

In the abstract: “hho1Δ (lacking the HHO1 gene, coding for the linker histone H1)”

 In the Introduction, more information was included about the two chromatin proteins, the linker histone H1, Hho1p, and the actin-related protein Arp4p. Please see Lines 82-93.

“In our previous studies, we have proven that the physical interaction between two crucial chromatin components – the yeast linker histone Hho1p, responsible for nucleosome stabilization and higher order chromatin structure maintenance [29-31], and the actin-related protein Arp4p – a subunit of Ino80, SWR1 and NuA4 [32], was of utmost importance for both yeast replicative and chronological ageing. The studied chromatin mutants were arp4 (with a point mutation in the ARP4 gene, coding for actin-related protein 4 - Arp4p), hho1Δ (lacking the HHO1 gene, coding for the linker histone H1), and the double mutant arp4 hho1Δ cells with the two mutations. We have shown that the disruption of the interaction between the linker histone and Arp4p affected the organization of chromatin structure, cellular morphology, and how cells responded to stress. Moreover, the double mutant cells that experienced premature ageing phenotypes had a reduced chronological and replicative lifespan and lowered replicative potential [33-37].”

P14, Chapter 3.5 does not add new understanding to the previous chapter. Last paragraph of the chapter is not the conclusion. Should be replaced. – changed:

Chapter 3.5. was connected to chapter 3.3. as subsection 3.3.2.

The last paragraph of the chapter was removed.  

P16 Chapter 3.6. Why 3.6 is separate chapter? Should be connected to the next chapter. – corrected:

Ex-chapter 3.6 is now chapter 3.4. and ex-chapters 3.7. and 3.8. are now connected to chapter 3.4. as subsections 3.4.1. and 3.4.2., respectively.

P17, L567 genes in italics – corrected: hho1Δ to hho1Δ

P17, L568, different than what? – corrected: “different than the wild type and the other two mutants”

P18, L588 S. cerevisiae, in italics – corrected: S. cerevisiae to S. cerevisiae

P19, L625, In hho1Δ strain – corrected: “In hho1Δ strain” instead of “In hho1Δ group”

P20, L653, crucially crucial, is too much - corrected: “crucially crucial for” to “crucial for”

P20, L668 genes in italics – corrected to “ATG18 and CDC28 transcripts in the arp4 hho1Δ mutant”

P20 L672 has Atg18 other roles in addition to this in autophagy process? – Yes

  1. cerevisiae Atg18p, as a subunit of the phosphatidylinositol 3-kinase complex that binds phosphatidylinositol-3,5-bisphosphate and ubiquitin, is involved in cytoplasm-to-vacuole transport by the CVT pathway as well as in late endosome to vacuole transport. In addition to its role in autophagy processes (autophagy of peroxisome, late nucleophagy; piecemeal microautophagy of the nucleus) Atg18 is involved in the vacuolar protein processing and the regulation of vacuole organization.

In the body text, chapter 3.5., Lines  650 we added:

“In yeast, as in higher eukaryotes, the nuclear-encoded protein Atg18 is a component of the Atg9•Atg2-Atg18 complex, which is crucial for autophagosome formation [107,108]. In addition, Atg18 participates in vacuolar morphology regulation, involving binding by phosphatidylinositol 3,5-bisphosphate [109].”

P20, L676 What encodes CDC28 gene? What is a function of Cdc28?.

Information must be given in Introduction.

ScCDC28 gene encodes the Cdk1, a protein serine/threonine kinase catalytic subunit of multiple cyclin-dependent kinase (CDK) complexes. By sequentially binding to G1, S, G2/M phase cyclins, Cdc28p acts as a master regulator of mitotic and meiotic cell cycles. It is involved in the regulation of many other cellular processes e.g., basal transcription, chromosome dynamics, DNA double-strand break processing and repair, protein localization to chromatin, metabolism, growth and morphogenesis etc.

In the body text, chapter 3.5., Line 643, we included:

“One of these genes is CDC28, which regulates the meiotic and mitotic cell cycle and is associated with G2/M checkpoint cell cycle blocking [106]. Additionally, it is part of cell growth regulation, metabolism, maintenance of chromatin dynamics and morphogenesis [106]. In particular, it was found that S. cerevisiae cdc28 mutants displayed reduced frequency of induced and spontaneous rho mutations and increased mitochondrial genome stability [103,104]. Damaged or superfluous mitochondria are removed using the autophagic machinery in the selective process of mitophagy.”

P21, L698, histone level. Which one? -  corrected:

“[91]. For example, chromatin structure has been found to influence mtDNA content as improper chromatin assembly or reduction in histone H3 and H4 levels in S. cerevisiae has been shown to promote mtDNA copy number increment [27].”

P21, L701, What is CRCs? - CRCs stands for Chromatin Remodeling Complexes.

In the text is corrected: “ATP-dependent Chromatin Remodelling Complexes (CRCs;…)”

P21, L703, Starting new paragraph “on the other hand” again? Is in line 699.  – corrected: “On the other hand” is removed.

P21, L719 , nine? – the text was revised

Round 2

Reviewer 1 Report

The authors have addressed my concerns.

Author Response

Dear reviewer,

Thank you for all your remarks and comments.
They did indeed help to improve our work.

Warm regards,
Milena

Author Response

Dear Reviewer,

Thank you very much indeed for these corrections.

We have amended all of them.

We want to thank you for the whole process of reviewing our work. All comments and remarks helped us improve the quality of it, and we shall be happy if it is published in JoF.

Warm regards,
Milena

PS. See all our edits according to the provided feedback:

√ P8, L255,  arp4 hho1Δ. Genes in italics - italicised

√ P9, L272, COX1, COB, 21rRNA, genes in italics - italicised

√ P11, L345, Interestingly. Do not start the sentence with “and”. - corrected

√ P12, L425, COX1. Genes in italics - italicised

√ P12, L431, hho1Δ. Genes in italics - italicised

√ P13, L470, arp4. Italics - italicised

√ P15, L556, arp4. Italics - italicised

√ P16, L596, rho zero or rho minus?

According to the authors Zubko, E.I. and Zubko, M.K. 2014 [92], the ethidium bromide-induced respiratory mutants of “…all six species had deletions in analysed mtDNA sequences…”. They suggested that the studied S. cerevisiae strains were most likely rho0 mutants (“…most likely, they were rho0 mutants.”). However, the complete lack of mtDNA has not been unequivocally proven, and in the body text, the authors themselves do not specify that the mutants are rho0, instead, they use the most common term, ‘rho mutants’. Therefore, we consider using “rho mutants” more correct than rho0.

[92] Zubko EI, Zubko MK. Deficiencies in mitochondrial DNA compromise the survival of yeast cells at critically high temperatures. Microbiological Research. 2014, 169( 2–3):185-195, https://doi.org/10.1016/j.micres.2013.06.011

√ P18, L648, rho zero or rho minus?

Devin and co-authors [ref. 103] reported reduced spontaneous and induced mitochondrial rho- mutability in S. cerevisiae cdc28 mutants, while Chen et al. [104] investigated rho° mutants. To summarize the two studies, the term ‘rho mutants’ is used in the present manuscript.

[103] Devin AB, Prosvirova TYu, Peshekhonov VT, Chepurnaya OV, Smirnova ME, Koltovaya NA, Troitskaya EN, Arman IP. The start gene CDC28 and the genetic stability of yeast. Yeast. 1990, 6(3):231-43. doi: 10.1002/yea.320060308. PMID: 2190433.

[104] Chen S, Liu D, Finley RL Jr, Greenberg ML. Loss of mitochondrial DNA in the yeast cardiolipin synthase crd1 mutant leads to up-regulation of the protein kinase Swe1p that regulates the G2/M transition. J Biol Chem. 2010, 285(14):10397-407. doi: 10.1074/jbc.M110.100784.

√ P18, L658, ATG18 and CDC28. Genes in italics - italicised

√ P18, L666- 667, Genes in italics - italicised

√ P20, L748-749, Genes in italics - italicised

√ References, Saccharomyces cerevisiae and genes in italics.

References №16, 28, 29, 36, 38, 39, 42, 48, 50, 54, 59, 71, 76, 78, 79, 99, 100, 101, 103, 105, 108 & 110 species names and gene names are italicised. For References №32, 40 & 106 – EndNote does not accept the correction.